# Synthesis of Enantiopure (*S*)-Atenolol by Utilization of Lipase-Catalyzed Kinetic Resolution of a Key Intermediate

**DOI:** 10.3390/ijms25063497

**Published:** 2024-03-20

**Authors:** Mari Bergan Hansen, Anna Lifen Tennfjord, Fredrik Heen Blindheim, Lucas Hugo Yvan Bocquin, Elisabeth Egholm Jacobsen

**Affiliations:** Department of Chemistry, Norwegian University of Science and Technology (NTNU), Høgskoleringen 5, N-7491 Trondheim, Norway; mari.bergan@live.no (M.B.H.); anna.tennfjord@gmail.com (A.L.T.); fredrik.heen.blindheim@rise-pfi.no (F.H.B.); lucas.bocquin@uni-bielefeld.de (L.H.Y.B.)

**Keywords:** enantiopure (*S*)-atenolol, *Candida antarctica* lipase B, base catalysis, Chiralcel OD-H column

## Abstract

(*S*)-Atenolol ((*S*)-2-(4-(2-Hydroxy-3-(isopropylamino)propoxy)phenyl)acetamide) has been synthesized in >99% enantiomeric excess (*ee*) with the use of *Candida antarctica* lipase B from Syncozymes (Shanghai, China), in a kinetic resolution of the corresponding racemic chlorohydrin. A catalytic amount of base was used in deprotonation of the phenol building block. The enantiopurity of the chlorohydrin building block remained unchanged upon subsequent amination to yield the final drug. All four steps in the synthesis protocol have been optimized compared to previously reported methods, which makes this new protocol more sustainable and in accordance with green chemistry principles. The overall yield of (*S*)-atenolol was 9.9%, which will be further optimized.

## 1. Introduction

The American Heart Association reported in March 2019 their update on heart diseases and stroke statistics. The report states that high blood pressure concerned 46% of the total population at ages 20 years and older in the United States between 2013–2016. It was the cause of death for 82,735 Americans in 2016 and cost the American society approximately USD 55.9 billion in the period 2014–2015 [1]. In 2018, cardiovascular treatment made up 4.9% of the total pharmaceutical market in Norway, which corresponded to NOK 1.17 million [2]. A class of drugs that have been used in the treatment of both cardiovascular and non-cardiovascular diseases are the β-adrenergic blocking agents (beta-blockers). Approximately 300,000 patients in Norway use beta-blockers [3]. Worldwide, the use of beta-blockers increases year by year and the sales are estimated to account for USD 13,684 million by 2030 [4].

The highly polar cardio selective β1-antagonist atenolol is selective towards β1-receptors found in the heart. This drug is used in the treatment of hypertension, angina pectoris and arrythmia. Atenolol is one of the most widely used beta-blockers clinically and is often used as a reference drug for comparisons with other antihypertensives. The drug might, however, be even more effective in preventing myocardial infarction [5]. 

Atenolol is manufactured with enantiomerically pure active pharmaceutical ingredient (API) as Atpure^®^ by Emcure Pharmaceuticals (Pune, India), and with racemic API under the names Tenormin^®^, Mylan^®^ and others. The eudismic ratio is 46 in favour of (*S*)- to (*R*)-atenolol [6], and studies in rats show that the *R*-enantiomer has no effect. While the racemic drug causes a lowering of the heart rate, this side effect is not observed with enantiopure (*S*)-atenolol [7].

Several synthesis protocols for producing enantiopure (*S*)-atenolol have been published. Emcure Pharmaceuticals uses enantiopure epichlorohydrin to produce enantiopure (*S*)-atenolol from deprotonated 2-(4-hydroxyphenyl)acetamide. The specific rotation of the final drug (*S*)-atenolol of ${\left[\alpha \right]}_{D}^{25}$ = −17.1 (1.0, 1N HCl) is reported [8]. Dwivedee et al. also started with deprotonation of 2-(4-hydroxyphenyl)acetamide gaining only the epoxide 2-(4-(2-oxiran-2-ylmethoxy)phenyl)benzeneacetamide. Reaction of this epoxide with acetyl chloride in methanol gave the racemic 4-(3-chloro-2 hydroxypropoxy)benzeneacetamide, which was resolved by several lipase preparations and vinyl acetate as the acyl donor. The authors claim to have formed (*S*)-atenolol from (*S*)-4-(3-chloro-2-hydroxypropoxy)benzeneacetamide [9]. This is not possible according to their reaction conditions, since addition of isopropylamine in water to (*S*)-4-(3-chloro-2-hydroxypropoxy)benzeneacetamide leads to (*R*)-atenolol. The authors do not report any specific rotation of their (claimed) (*S*)-atenolol product, nor of their enantiopure building blocks. Agustian et al. present a similar study of (*S*)-atenolol synthesis using several lipase preparations to resolve the racemic 4-(3-chloro-2 hydroxypropoxy)benzeneacetamide [10]. The authors present no evidence of the absolute configuration of the product, nor any yields or evidence of the enantiomeric excess (*ee*) of the product. Sikora et al. reported in 2020 the kinetic resolution of racemic atenolol catalyzed by lipase from *Candida rugosa* with isopropenyl acetate as the acyl donor, resulting in the acetate of (*S*)-atenolol in 94% *ee*. The authors present no evidence of the absolute configuration of this acetate, nor of the unreacted (*R*)-atenolol [11,12]. The authors have previously published several articles on lipase catalyzed kinetic resolution of racemic atenolol with the amide of (*S*)-atenolol presented as the acetate [13,14,15]. (*S*)-Atenolol has been synthesized in 98% *ee* in a seven-step method using Jacobsen’s catalyst ((*R*,*R*)-salen Co(III)OAc) [16], and in 94% *ee* through kinetic resolution of racemic atenolol using lipase from *Pseudomonas cepacia* [17]. We have produced the enantiopure building block (*R*)-4-(3-chloro-2 hydroxypropoxy)benzeneacetamide starting with a deprotonation of 2-(4-hydroxyphenyl)acetamide with sodium hydroxide, and by using lipase B from *Candida antarctica* (CALB) in the kinetic resolution of the racemic chlorohydrin 4-(3-chloro-2 hydroxypropoxy)benzeneacetamide, the enantiopure chlorohydrin was obtained with 99% *ee* [18]. We have now improved the yield of the building block and reduced the amounts of reactants used. The enantiopure drug (*S*)-atenolol has been synthesized from the enantiopure chlorohydrin.

## 2. Results and Discussion

### 2.1. Addition of Epichlorohydrin to 2-(4-hydroxyphenyl)acetamide (***1***)

The first step in the synthesis of (*S*)-atenolol ((*S*)-**4**) is the formation of 4-(3-chloro-2-hydroxypropoxy)benzeneacetamide (**2a**) and 2-(4-(2-oxiran-2-ylmethoxy)phenyl)benzeneacetamide (**2b**) from phenol 2-(4-hydroxyphenyl)acetamide (**1**), sodium hydroxide and epichlorohydrin. The impact of the amount of sodium hydroxide, epichlorohydrin and acetic acid used in the reaction was investigated in order to improve the overall yield of the product compared to earlier reported data (Figure 1).

When stoichiometric amounts of sodium hydroxide were used to deprotonate the phenolic proton of **1**, a small peak at t_R_ = 13.37 min was seen in the HPLC chromatogram together with the desired product chlorohydrin **2a** (t_R_ = 12.06 min) and the epoxide **2b** (t_R_ = 9.58 min), see Figure 1. LC-MS analysis of the reaction mixture on an AQUITY UPLC BEH C18-column with an isocratic mobile phase composition of water and acetonitrile and (30:70) with 1% formic acid and a flow of 0.2 mL/min showed a peak with the molecular mass of 382.04 g/mol, which corresponds to the mass of C_19_H_21_N_2_O_5_Na (Figure 2). The compound then has a molecular mass of 358.39 g/mol, which corresponds to compound **2c**. We have previously predicted the mechanism of the formation of this dimeric ether compound in syntheses of similar beta-blockers. Here, we also present full characterization of the corresponding by-product in the synthesis of enantiopure (*S*)-esmolol [19].

When two equivalents of epichlorohydrin were added to a solution containing phenol **1** and catalytical amounts of sodium hydroxide, the by-product **2c** was not observed after 4 h of reaction. A proposed mechanism for the reaction of **1** to **2a** and **2b** with the regeneration of the base is shown in Figure 2. Phenol **1** is deprotonated to form phenoxide **1′** which can attack epichlorohydrin at carbon 1 or 3 (reaction mechanism a and b, respectively) resulting in phenoxide **2a′** and the epoxide **2b**. Phenoxide **2a′** may react in two ways: by deprotonating a water molecule forming chlorohydrin **2a,** thus regenerating the base (reaction mechanism c), or by an intramolecular S_N_2-like reaction forming epoxide **2b** (reaction mechanism d). HPLC analyses of the product mixtures from the reaction of **1** to **2a** and **2b** showed that the use of catalytic amounts of base favoured the formation of **2a** over the epoxide **2b**. When one or two equivalents of base were used relative to **1**, **2b** and **2c** were predominant after 6 h of reaction time.

The use of catalytic amounts of base is possible due to the regeneration of the base during the deprotonation of water during the formation of **2a** from **2a′**. For a similar reason, it is also possible to use less acetic acid than previously reported in the syntheses of other beta-blockers. We have previously used between five and ten equivalents of acetic acid and two to five equivalents of lithium chloride [20]. By addition of excess lithium chloride relative to **2b** in the reaction mixture and approximately five equivalents of acetic acid relative to **2b**, a 52% yield of **2a** was achieved. Since the molar masses of **2a** and **2b** are different and the HPLC analyses yield only relative amounts of each compound in the reaction mixture, the amounts of the reactants are estimated.

### 2.2. Lipase Catalyzed Kinetic Resolution of Chlorohydrin ***2a***

A CALB catalyzed kinetic resolution of **2a** (separation of the enantiomers of **2a** on the Chiralcel column is shown in Appendix A) with vinyl butanoate as the acyl donor in acetonitrile produced (*R*)-**2a** in 99% *ee* with 32% yield (Figure 1, Appendix A). Appendix A and Figure 3 show the chromatograms of the reaction after 1, 5 and 24 h of a total of 27 h reaction time. After 27 h, no (*S*)-**2a** was seen in the chromatogram and the *ee* of (*R*)-**2a** was >99.0% 

The column used did not allow sufficient separation of the two enantiomers of the butanoic ester (*S/R*)-**3** to measure the enantiopurity of the ester. In order to verify the enantiomer formed, the ester was hydrolyzed and identified as the (*S*)-**3** enantiomer. We have previously obtained a yield of 16% of (*R*)-**2a**, and the *E*-value of the kinetic resolution of **2a** was >200. In our previous article, we also presented data of the *S*-enantiomer of the chlorohydrin, (*S*)-**2a** [18].

### 2.3. Synthesis of (S)-Atenolol, (S)-***4***

The drug (*S*)-**4** was synthesized in 60% yield and >99% *ee* by amination of (*R*)-**2a** (*ee* > 99%) with isopropylamine in water reacting at room temperature for 48 h (Figure 1). The specific rotation was ${\left[\alpha \right]}_{D}^{23}$ = −17.0 (1.0, 1N HCl), which is in accordance with previously reported data [8]. The overall yield of *(S)-**4*** is 9.9%, which will be further improved.

### 2.4. Synthesis of Racemic Atenolol (***4***) and Attempt to Resolve ***4*** with CALA

Another attempt to synthesizing enantiopure (*S*)-atenolol was performed in a kinetic resolution of racemic atenolol (**4**) by use of lipase A from *Candida antarctica*. Racemic **4** was produced directly from the reaction mixture of **2a** and **2b** (without the ring opening of **2b** with lithium chloride) with addition of isopropylamine in methanol resulting in **4** in 42% yield (Figure 3).

Attempts to resolve **4** in acetonitrile with lipase A from *Candida antarctica* (CALA) as the catalyst and vinyl butanoate as the acyl donor were performed with an *E*-value of 1.8 (Figure 4). We have previously had success with using CALA as the catalyst in the kinetic resolution of secondary alcohols with two large groups connected to the stereo centre, which atenolol also has [21]. But the amine group in **4** seems to be the problem, since the amide (s) **4d** may be formed in addition to the ester (s) **4b**, see the chromatogram from the reaction in Figure 4.

The *ee* of the unreacted (*S*)-atenolol was 4% (*ee*_s_ ((*S*)-atenolol, (*S*)-**4**)) and the *ee* for the product ester (*R*)-**4b** was 27% (*ee*_p_). The retention times were assigned due to the known stereo preference of CALA [21], t_R_ (*S*)-**4** = 18.36 min and t_R_ (*R*)-**4** = 28.97 min, R_S_ = 2.72, t_R_ (*S*)-**4b** = 25.04 min and t_R_ (*R*)-**4b** = 33.87 min, Chiralcel OD-H column with a gradient mobile phase *i*-PrOH/*n*-hexane; 9:95 (0 min)–10:90 (10 min)–60:40 (80 min), flow 0.5 mL/min (Figure 4).

These results show that CALA exhibits low selectivity for the enantiomers of atenolol (**4**). The racemic compound with t_R_ = 14.15 min and t_R_ = 16.94 min is anticipated to be the enantiomers of the amide product **4d** (Figure 4). Chałupka (Sikora) et al. claim that they have succeeded in resolving racemic atenolol with *Candida rugosa* lipase with vinyl acetate as the acyl donor and ionic liquids/toluene as the solvent yielding 94% *ee* of the (*S*)-atenolol acetate [11]. However, the absolute configuration of this product has not been confirmed.

## 3. Materials and Methods

### 3.1. Chemicals

All chemicals are commercially available and of analytic grade. The chemicals were bought from Sigma-Aldrich Norway (Oslo, Norway). HPLC grade solvents were used for HPLC analyses. Dry solvents (tetrahydrofuran and acetonitrile) were prepared with a solvent purifier, MBraun MDSPS800 (München, Germany). *n*-Hexane was dried manually by adding molecular sieves (4Å) to the solvent 24 h before use. Molecular sieves (1/8 pellets, pore diameter 4 Å) were placed in a porcelain dish and dried at 1000 °C for 24 h and kept in a desiccator thereafter.

### 3.2. Enzymes

*Candida antarctica* Lipase B (CALB) (activity ≥ 10,000 PLU/g, 1 unit corresponds to the synthesis of 1 mmol per minute propyl laureate from lauric acid and 1-propanol at 60 °C, lot no. 20170315), immobilized at high hydrophobic macroporous resin, produced in fermentation with genetically modified *Pichia pastoris* was gifted from Syncozymes Co, Ltd. (Shanghai, China). *Candida antarctica* lipase A (activity = 725 U/g, lot no. VZ1030-12, batch no 080116) immobilized on microporous beads was a gift from Viazym BV (Delft, The Netherlands).

The enzymatic reactions were performed in a New Brunswick G24 Environmental Incubator Shaker from New Brunswick Co. Inc. (Edison, NJ, USA) or in an Infors Minitron (Infors AG, Bottmingen, Switzerland).

### 3.3. General Analyses

TLC was performed on Merck silica 60 F_254_ and detected by UV at λ = 254 nm. Flash chromatography was performed on silica gel from Sigma-Aldrich (Oslo, Norway). Pore size 60 Å, 230–400 mesh particle size, 40–63 μm particle size.

### 3.4. High-Performance Liquid Chromatography (HPLC)

Achiral HPLC analyses were performed on an Agilent 1290 system equipped with an auto injector (4 μL), and detection was performed by a diode array detector (DAD, l = 254 nm) (Santa Clara, CA, USA). All separations of **1, 2a** and **2b** were performed on an Agilent Zorbax Eclipse XBD-C18 column (150 mm L × 4.6 mm i.d., 5 μm particle size) from Matriks (Oslo, Norway) with an isocratic eluent (H_2_O:MeCN, 75:25) over 5 min with a flow of 1.0 mL/min, which produced t_R_ **1b** = 1.86 min, t_R_ **2b** = 3.04 min and t_R_
**2a** = 3.32 min. When stoichiometric amount of sodium hydroxide was used also, the by-product **2c** was seen, then a linear gradient mobile phase composition of H_2_O and MeCN (75:25)–(65:35) over 20 min with 0.5 mL/min flow was the method on the Zorbax Eclipse XDB C18-column, producing t_R_ **1** = 3.03 min, t_R_ **2a** = 12.06 min, t_R_ **2b** = 9.59 min and the by-product t_R_ **2c** = 13.38 min.

LC-MS analysis of **2c** was performed on an ACQUITY UPLC BEH C18 column (100 mm L × 2.1 mm i.d., 1.7 μm particle size) from Waters^TM^ (Waters Norway, Oslo, Norway) with isocratic mobile phase (H_2_O:MeCN, 30:70) with 1% formic acid and a flow of 0.2 mL/min resulting in a mass of 382.04 g/mol which corresponds to C_19_H_21_N_2_O_5_Na. The calculated mass of **2c** is 358.39 g/mol, the formula for which is C_19_H_22_N_2_O_5_.

Chiral HPLC analyses were performed on an Agilent HPLC 1100 with a manual injector (Rheodyne 77245i/Agilent 10 μL loop) (Santa Clara, CA, USA). A Chiralcel OD-H column from Daicel, Chiral Technologies Europe (Gonthier d´Andernach, Illkirch, France) was used (250 mm L × 4.6 mm i.d., 5 μm particle size) in addition to the reverse phase corresponding column Chiralcel OD-RH (150 × 4.6 mm i.d., 5 μm particle size). The method used for all analyses was *i*-PrOH:*n*-hexane, 17:83, flow 1 mL min^−1^, UV 254 nm. The enantiomers of **2a** eluted with t_R_ (*S*)-**2a** = 41.52 min, t_R_ (*R*)-**2a** = 46.32 min, R_S_ = 1.74. The purified (*R*)-**2a** was analyzed by the same method as racemic **2a**: t_R_ (*R*)-**2a** = 46.32 min, with no presence of (*S*)-**2a** in the chromatogram, resulting in an enantiomeric excess of >99% *ee*. For determination of the *E*-value of the enzyme catalyzed kinetic resolution of **2a**, see Lund et al. [18]. With the use of Chiralcel OD-RH (reverse phase), the retention times were t_R_ (*R*)-**2a** = 6.32 min and t_R_ (*S*)-**2a** = 7.17 min. Retention times of **4** on Chiralcel OD-H: t_R_ (*S*)-**4** = 18.37 min and t_R_ (*R*)-**4** = 28.98 min, R_S_ = 2.72. Purified (*S*)-atenolol: t_R_ (*S*)-**4** = 18.39 min.

### 3.5. NMR Analyses

NMR analyses were recorded on a Bruker 400 MHz Avance III HD instrument equipped with a 5 mm SmartProbe Z-gradient probe operating at 400 MHz for ^1^H and 100 MHz for ^13^C, respectively, or on a Bruker 600 MHz Avance III HD instrument equipped with a 5 mm cryogenic CP-TCI Z-gradient probe operating at 600 MHz for ^1^H and 150 MHz for ^13^C (Bruker, Rheinstetten, Germany). Chemical shifts are in ppm relative to TMS and coupling constants are in hertz (Hz).

### 3.6. Mass Spectroscopy Analyses

Accurate mass determination in positive and negative mode was performed on a ”Synapt G2-S” Q-TOF instrument from Waters™ (Waters Norway, Oslo, Norway). Samples were ionized by the use of ASAP probe (APCI). Calculated exact mass and spectra processing were performed by Waters™ Software (Masslynxs V4.1 SCN871).

### 3.7. Infrared Spectroscopy Analyses

Infrared spectroscopy was performed on a NEXUS FT-IR model 470 instrument from Thermo Nicolet Corporation (Madison, WI, USA).

### 3.8. Specific Rotation Analyses

Specific rotation was determined on a PerkinElmer Model 341 Polarimeter (Waltham, MA, USA), with a cell of 10 cm length, λ 589 nm.

### 3.9. Assignment of Absolute Configurations

Absolute configuration of the faster reacting enantiomer in lipase catalyzed resolution was determined by the known enantioselectivity of CALA [21] and CALB [22] and by comparing the elution orders of the atenolol building block enantiomers and drug enantiomer with previous analyses of similar α-halogenated 1-(4-benzyloxy)phenyl)ethanols on the Daicel Chiralcel OD-H column. In general, the *R*-enantiomers are the most retained [23].

### 3.10. Synthesis Protocols

#### 3.10.1. Synthesis of Chlorohydrin **2a** and Epoxide **2b**

To an aqueous solution of NaOH (0.2M, 8.71 mL, 1.98 mmol) and 2-(4-hydroxyphenyl)-acetamide (**1**) (1.00 g, 6.62 mmol), epichlorohydrin (1.04 mL, 13.26 mmol) was added and set to stir at rt. for 4 h. The reaction was monitored by TLC (MeOH:CH_2_Cl_2_, 1:5): R_f_ **2a** = 0.56, R_f_ **2b** = 0.63. The reaction mixture was washed with H_2_O in vacuo producing a 1.44 g mixture of chlorohydrin **2a** and epoxide **2b** as a white solid, analyzed on the Zorbax Eclipse XBD-C18 HPLC column with an isocratic mobile phase (MeCN:H_2_O, 25:75) over 5 min, flow = 1.0 mL/min, t_R_ (**1**) = 1.86 min, t_R_
**2a** =3.32 min and t_R_ **2b** = 3.04 min.

#### 3.10.2. Synthesis of Chlorohydrin **2a** by Addition of Lithium Chloride and Acetic Acid to Epoxide **2b**

To the mixture of chlorohydrin **2a** and epoxide **2b** (1.44 g), MeCN (20 mL), LiCl (0.28 g, 6.6 mmol) and AcOH (1.98 mL, 34.62 mmol) were added and stirred at rt. for 24 h. The reaction was monitored by TLC (CH_2_Cl_2_:MeOH, 1:5), R_f_
**2a** = 0.56. (See Appendix A). The reaction mixture was quenched with Na_2_CO_3_ (pH 12) to a neutral pH followed by extraction with CH_2_Cl_2_ (3 × 50 mL). The organic layer was dried over MgSO_4_ and the solvent was removed under reduced pressure and further in vacuo producing chlorohydrin **2a** in 52% yield (0.85 g, 3.49 mmol) as a white solid; mp. 143–145 °C. The conversion of epoxide **2b** was analyzed on the Zorbax Eclipse XBD-C18 HPLC column with an isocratic mobile phase (MeCN:H_2_O, 25:75) over 5 min, flow = 1.0 mL/min, t_R_ **2a** =3.32 min. ^1^H NMR (600 MHz, DMSO-*d_6_*) δ ppm: 7.38 (s, 1H, -NH-H), 7.17–7.16 (m, 2H aromatic), 6.88–6.86 (m, 2H, aromatic), 6.82 (s, 1H, -NH-H), 5.54–5.53 (d, 1H, *^3^J*_HH_ = 5.13 Hz, -OH), 4.04–4.00 (sextet, 1H, *^3^J*_HH_ = 5.13 Hz, -CH-OH), 3.97–3.92 (m, 2H, -O-CH_2_-), 3.68–3.65 (2H, m, 2H, CH_2_-Cl), 3.28 (s, 2H, -CH_2_-CONH_2_); ^13^C NMR (600 MHz, DMSO-*d_6_*) δ ppm: 172.9, 157.5, 130.5 (2C), 129.2, 114.7 (2C), 69.5, 69.1, 47.2, 41.8. MS (TOF-ASAP): [M + H]^+^ 244.0739, (calcd. C_11_H_14_NO_3_Cl, 243.686761). IR (cm^−1^): 3349, 1633, 1241, 706. 

#### 3.10.3. Synthesis of Racemic Atenolol (4) Directly from Phenol **1**

2-(4-Hydroxyphenyl)acetamide (**1**) (2.52 g, 16.67 mmol) was stirred in 2-(chloromethyl)oxirane (epichlorohydrin) (13 mL, 165.80 mmol) at rt., and was added a solution of NaOH (0.33 g, 8.25 mmol) in H_2_O (5 mL). After 48 h, full conversion was observed by TLC (MeOH:CH_2_Cl_2_: 1:4), which showed the presence of both **2a** and **2b**. The mixture was filtered, and the solids were dried under reduced pressure, before the crude product was dissolved in MeOH (25 mL) and added *i*-PrNH_2_ (10 mL, 116.39 mmol). After stirring for 24 h, full conversion of **1** to **4** was observed by TLC. The solvent was removed under reduced pressure. This afforded 5.49 g crude product of which 1.03 g was recrystallized from MeCN, afforded atenolol (**4**) in 42% total yield (0.35 g, 1.31 mmol) and >98% purity (NMR). ^1^H NMR (400 MHz, CD_3_OD) δ ppm: 7.26–7.24 (d, 2H, aromatic, *^3^J*_HH_ = 7.5 Hz), 6.95–6.93 (d, 2H, aromatic, *^3^J*_HH_ = 7.5 Hz), 4.24–4.22 (m, 1H, CH-OH, *^3^J*_HH_ = 4.4 Hz), 4.08–3.99 (m, 2H, -O-CH_2_-, *^3^J*_HH_ = 5.0 Hz, 9.5 Hz), 3.49–3.43 (m, 3H, CO-CH_2_-, -NH-CH-, *^3^J*_HH_ = 6.6 Hz), 3.30–3.27 (m, 1H, -CH_2_-NH-, *^3^J*_HH_ = 13.2 Hz), 3.18–3.12 (t, 1H, -CH_2_-NH-, *^3^J*_HH_ = 10.5 Hz), 1.39–1.38 (m, 6H, *^3^J*_HH_ = 3.5 Hz). ^13^C NMR (400 MHz, CD_3_OD) δ ppm: 175.8, 157.5, 129.9, 128.4, 114.3, 69.6, 65.6, 50.6, 47.1, 41.0, 18.0, 17.4.

#### 3.10.4. Synthesis of (*R*)-2-(4-(3-chloro-2 hydroxypropoxy)phenyl)acetamide, (*R*)-**2a**

Chlorohydrin **2a** (0.56 g, 2.3 mmol) and vinyl butanoate (1.43 g, 12.5 mmol) were added to a flask with dry MeCN (40 mL) and molecular sieves. CALB (0.71 g) was added, and the reaction was incubated at 30 °C and 200 rpm for 27 h in an incubator shaker. The enzyme and the molecular sieves were filtered off and the solvent was removed under reduced pressure. The ester (*S*)-**3** and the chlorohydrin (*R*)-**2a** were separated on a silica column with EtOAc as eluent. This yielded (*R*)-**2a** (0.090 g, 0.37 mmol, 32% yield), *ee* > 99%. ${\left[\alpha \right]}_{D}^{23}$ = −3.0 (1.0, MeOH), which is in accordance with our previous reported data of (*R*)-**2a** [18]. The *E*-values and K_eq_ were calculated by the software program E&K Calculator 2.1b0 PPC [24].

#### 3.10.5. Synthesis of (*S*)-Atenolol, (*S*)-**4**

To (*R*)-**2a** (0.090 g, 0.37 mmol), *i*-PrNH_2_ (3 mL, 34.9 mmol) and dist. H_2_O (1.0 mL) were added. The reaction was stirred at rt. for 48 h until TLC (MeOH:CH_2_Cl_2_, 1:5) showed full conversion. This gave (*S*)-**4** as a white powder (0.054 g, 0.22 mmol, 60% yield), 99% purity (NMR), *ee* > 99%. ${\left[\alpha \right]}_{D}^{23}$ = −17.0 (1.0, 1N HCl). NMR spectra as for racemic **4**.

#### 3.10.6. Kinetic Resolution of 2-(4-(2-Hydroxy-3-(isopropylamino)propoxy)phenyl)acetamide, **4**

To a solution of 2-(4-(2-hydroxy-3-(isopropylamino)propoxy)phenyl)acetamide (**4**) (35.1 mg, 0.13 mmol) and vinyl butanoate (81.3 mg, 0.71 mmol) in MeCN (3 mL) placed in an incubator shaker at 30 °C and 200 rpm, CALA (20.4 mg) was added. Samples were collected over four days, which were analyzed by chiral HPLC (*n*-hexane:*i*-PrNH_2_:Et_2_NH: 80:20:0.1) which showed low enantioselectivity (*E* = 1.8) and the reaction was therefore not further analyzed.

## 4. Conclusions

A four-step synthesis of (*S*)-atenolol ((*S*)-**4**) in >99% *ee* has been performed, starting from 2-(4-hydroxyphenyl)-acetamide (**1**). The base catalyzed deprotonation of the starting material avoided formation of the by-product (**2c**), thus resulting in an overall yield of the racemic chlorohydrin **2a** to 52%. This is an improvement from our previous reported yield of **2a** of 22%. CALB catalyzed kinetic resolution of **2a** gave the enantiopure (*R*)-**2a** in >99% *ee* with 32% yield. CALB catalyzed kinetic resolution of chlorohydrin **2a** is an efficient method to obtain enantiopure building blocks for beta-blockers; however, the yield of this enzymatic step is limited to 50%. Our new protocol to achieve enantiopure (*S*)-atenolol has fewer steps and uses reduced amounts of all reagents in the synthesis compared to previous published protocols. The overall yield of (*S*)-**4a** of approx. 10% may be increased by improved work-up procedures.

## Data Availability

The data presented in this study are available in the article and Appendix A.

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
