# Peer review of "Synthesis of Enantiopure (S)-Atenolol by Utilization of Lipase-Catalyzed Kinetic Resolution of a Key Intermediate"

_ijms, 2024, doi:10.3390/ijms25063497_

Round 1

Reviewer 1 Report

Comments and Suggestions for Authors

Efficient methods that lead to formation of enantiomerically pure (S)-Atenolol can be of interest to the pharmaceutical industry and academia. The authors proposed a four-step synthesis of the target compound from readily available starting material.

The first step in synthesis based on the O-alkylation of 2-(4-hydroxyphenyl)acetamide (1) with epichlorohydrin led to the formation of a mixture of two products, chlorohydrin 2a and epoxide 2b. It’s not clear why the author describes this reaction as a ‘2.3 Base Catalyzed Deprotection of Phenol 1’.  This is very confusing to readers and cannot be accepted. The second step in the synthesis describes the synthesis of pure chlorohydrin 2a by treating the mixture obtained in first step. The experimental procedure includes lithium chloride (6.6 mmol) and acetic acid (34.62 mmol). The acid was used in large excess. The discussion on the reaction mechanism depicted in Scheme 2 did not corresponds to already used. Enantioselective enzymatic transesterification using vinyl butanoate, without optimisation, led to kinetic resolution reaction and unreacted alcohol was obtained with a yield of 32% only. The last step of ion synthesis N-alkylation of enantiomerically pure chlorohydrin 2a with isopropyl amine leads to the formation of enantiomerically pure (S)-Atenolol in 60% yield. The overall yield of synthesis 9.9 percent is very low and out of interest for industry.

In current form, the manuscript is full of mistakes and requires intensive corrections. The authors should reformulate their basic statements. The O-alkylation of the phenols reaction is definitely not deprotection of phenols reaction. The role of lithium ions in the second step reaction was highlighted while in the reaction mixture, a large excess of acetic acid was used. Simple additional experiments can explain the role of lithium chloride. The attempted stereoselective kinetic resolution of racemic 4 in n-hexane is a good starting point for the next experiments.      

Comments on the Quality of English Language

The explanation of the mechanisms of reaction is incorrect together with  nomenclature used. 

Author Response

See pdf file for our response to the reviewer 

Reviewer 2 Report

Comments and Suggestions for Authors

See attached

Comments on the Quality of English Language

Author Response

Please see pdf file for our response to the reviewer

Reviewer 3 Report

Comments and Suggestions for Authors

This manuscript describes a synthesis of the well-known drug Atenolol in an enantiopure form. The key novelty here is a kinetic resolution of an intermediate chlorohydrin, realized by lipase-catalysed acylation of the racemate. The authors have presented a good literature background and the experiments are adequately described. Although the novelty in this work is not great, it could be considered for publication after correction of some errors and inaccuracies. To start from the title – considering that the lipase catalyses only the kinetic resolution of one particular intermediate, I don’t think that this qualifies the synthesis as "Lipase catalyzed". The actual process that is catalyzed by the enzyme is not really a part of the synthetic sequence, it only removes the unwanted enantiomer. The base catalysis in the initial step on the other hand is too trivial and well-known to be underscored in the title. I would suggest reformulation of the title. One possibility is: "Synthesis of enantiopure (S)-Atenolol by utilization of Lipase-catalyzed kinetic resolution of a key intermediate." Of course, this is up to the authors to decide.

A number of other remarks and recommendations are included directly in the attached PDF. Please check carefully and revise your manuscript accordingly.

Comments on the Quality of English Language

The quality of English language is generally alright. Minor corrections may be needed.

Author Response

(The authors gave the same response as above.)
